# Water Quality and Mortality from Coronary Artery Disease in Sardinia: A Geospatial Analysis

**DOI:** 10.3390/nu13082858

**Published:** 2021-08-20

**Authors:** Maria Pina Dore, Guido Parodi, Michele Portoghese, Alessandra Errigo, Giovanni Mario Pes

**Affiliations:** 1Dipartimento di Scienze Mediche, Chirurgiche e Sperimentali, University of Sassari, Viale San Pietro 43, 07100 Sassari, Italy; mpdore@uniss.it (M.P.D.); parodiguido@gmail.com (G.P.); 2Department of Medicine, Baylor College of Medicine, One Baylor Plaza, Houston, TX 77030, USA; 3Cardiovascular Surgery Unit, AOU Sassari, Via Enrico de Nicola 14, 07100 Sassari, Italy; michele.portoghese@aousassari.it; 4Dipartimento di Scienze Biomediche, University of Sassari, Viale San Pietro 43, 07100 Sassari, Italy; a.errigo@studenti.uniss.it; 5Sardinia Longevity Blue Zone Observatory, 08040 Ogliastra, Italy

**Keywords:** mineral-rich water, hard water, geospatial analysis, IHD, Sardinia

## Abstract

The role of water hardness on human health is still debated, ranging from beneficial to harmful. Before the rise of drinking bottled water, it was a common habit to obtain supplies of drinking water directly from spring-fed public fountains. According to the geographic location, spring waters are characterized by a variable content of mineral components. In this ecological study, for the first time in Sardinia, Italy, the spatial association between spring water quality/composition and standardized mortality ratio (SMR) for coronary artery disease (CAD) in the decade from 1981 to 1991 was investigated using data retrieved from published databases. In a total of 377 municipalities, 9918 deaths due to CAD, including acute myocardial infarction (AMI), ICD-9 code 410, and ischemic heart disease (IHD), ICD-9 code 411–414, were retrieved. A conditional autoregressive model with spatially structured random effects for each municipality was used. The average SMR for CAD in municipalities with a predominantly “soft” (<30 mg/L) or “hard” (≥30 mg/L) water was, respectively, 121.4 ± 59.1 vs. 104.7 ± 38.2 (*p* = 0.025). More specifically, an inverse association was found between elevated calcium content in spring water and cardiovascular mortality (AMI: r = −0.123, *p* = 0.032; IHD: r = −0.146, *p* = 0.009) and borderline significance for magnesium (AMI: r = −0.131, *p* = 0.054; IHD: r = −0.138, *p* = 0.074) and bicarbonate (IHD: r = −0.126, *p* = 0.058), whereas weak positive correlations were detected for sodium and chloride. The lowest CAD mortality was observed in geographic areas (North-West: SMR 0.92; South-East: SMR 0.88), where calcium- and bicarbonate-rich mineral waters were consumed. Our results, within the limitation of an ecological study, confirm the beneficial role of waters with high content in calcium and bicarbonate against coronary artery disease.

## 1. Introduction

Water is an essential component of the human diet, and the impact of its mineral constituents on human health has frequently raised concerns [1]. However, although the association between the levels of specific minerals dissolved in drinking water and health outcomes has been the object of several systematic investigations, a perspective based on the global composition of water or the combination of its constituents has received less attention [2]. Beyond providing hydration, water is an important source of essential elements. One aspect that has been most debated in the literature is the relationship between total dissolved solids (TDS) in drinking water and long-term health outcomes, and more specifically, the influence of alkaline earth ions such as calcium and magnesium, i.e., the so-called water hardness, on human wellness. It has been known for over a century that the mortality rate in a population is related to the hardness of water, but it was only at the end of the 1950s that the first systematic studies aimed at clarifying this relationship were conducted [3]. Most studies have especially focused on the relationship between water quality and cardiovascular disease, as this is the leading cause of death in high-resource countries. It has been estimated 422.7 million cases and 17.9 million deaths every year, nearly 31% of all global deaths [4,5]. For many years, mineral-rich waters, defined as TDS exceeding the threshold of 1500 mg/L [6], have been considered more harmful than beneficial for human health [7,8,9,10]. For example, naturally saline waters, or those that became hypersaline as a result of seawater infiltration into the ground, were often blamed for increasing blood pressure in the consumers [10,11,12], thus raising their background cardiovascular risk [13]. More specifically, sodium content in water was considered deleterious for its potential to induce blood hypertension, although chloride concentration may contribute as well [8,14,15,16]. Nonetheless, other studies seem to indicate beneficial outcomes from mineral-rich water consumption, including the life-span extension in long-lived populations [17,18]. In reality, as far as the relationship between mineral-rich water and cardiovascular risk is concerned, although short-term clinical studies may display transient adverse effects, i.e., a pressure-raising effect, in comparison with epidemiological studies entailing long term water consumption, accumulating evidence consistently indicates that regular consumption of waters rich in calcium and magnesium, provides a health benefit, especially in terms of reducing cardiovascular disease burden [3,19]. More importantly, bicarbonate anion, which may be present at high concentration in alkaline waters, does not exhibit a blood pressure-raising effect [20], therefore, the consumption of waters where chloride is mainly replaced by bicarbonate seems not to be harmful and may even show a healthier profile [21]. In recent research, we showed that the consumption of mineral-rich water did not increase pressure values [22]. However, physicians keep on suggesting to avoid consumption of waters high in TDS to prevent hypertension and its sequelae without putting attention on distinguishing between those rich in sodium and chloride (potentially harmful) from those rich in calcium and magnesium (beneficial) [23].

Although tap water is generally safe, in many areas of the world, bottled mineral water is preferred for daily usage due to several reasons, including a better taste [24]. This is the case of the population living on the island of Sardinia, which in the last few decades became one of the Italian regions with the highest percentage of people consuming bottled water because of distrust of using tap water as a primary drinking source [25]. However, older Sardinians have spent the most part of their life consuming spring water from local sources, which are widespread throughout the island, especially in rural areas (Figure 1).

In addition, at that time, mortality for CAD was one of the lowest at the national level [26]. According to the complex composition of the bedrocks present in the island microplate, spring waters are characterized by a variable content of mineral components, ranging from low- to high-saline composition.

This scenario prompted us to investigate through an ecological study the spatial association between spring waters quality and the mortality rate for coronary artery disease in the Sardinia island.

## 2. Materials and Methods

### 2.1. Setting

The study was conducted in the entire Sardinian region, a Mediterranean island belonging to Italy. The island is about 180 km off the mainland coast with an area of 24,090 sq km and a population of 1.6 million subdivided into 4 provinces and a total of 377 municipalities. The geology of the island is complex and consists of a crystalline basement related to the European Variscan belt, in which igneous and metamorphic rocks, mainly consisting of granite and basalt, are surrounded by sedimentary rocks essentially composed of limestone and dolomite [27,28]. The hydrogeological constitution of the island is remarkably heterogenous and includes a central-northern mountainous site laying on a granite base of plutonic rocks whose waters, low in pH, are poor in minerals and interact with rocks during many years of flow in generally impermeable layers following the various fault systems that have occurred over the course of orogeny [27], whereas, the surrounding sedimentary rocks are especially rich in magnesium and calcium [29]. This peculiar petrographic configuration of the island results in a wide variety of spring water quality, making the island an ideal model for studying the relationships between water composition and population health indicators. Sardinia is a scarcely industrialized region, and a part of the population lives in small rural villages where agri-pastoral activities still prevail [30]. More importantly, Sardinians share a homogeneous genetic background [31].

### 2.2. Data Collection

Analyzed data were collected from all Sardinian municipalities, corresponding to 377 territorial units. Raw mortality data for CAD in the period ranging between 1981 and 1991 were retrieved from the “Atlante di Mortalità in Sardegna” (Sardinia Mortality Registry), [32]. The registry provided the number of observed and expected deaths for each disease per municipality according to the International Classification of Diseases 9th revision (ICD-9) and allowed the calculation of adjusted standardized mortality ratio (SMR). More specifically, data on acute myocardial infarction (AMI, ICD code 410) and other ischemic heart diseases (IHD, ICD codes 411–414 including acute and subacute forms of coronary artery disease (ICD code 411), previous myocardial infarction (ICD code 412), angina pectoris (ICD code 413), and additional forms of chronic IHD (ICD code 414), were selected. When a municipality had a number of deaths from CAD inferior to 1 unit, data from adjacent municipalities were clustered.

### 2.3. Water Composition

The physico-chemical parameters in the spring waters of the 377 Sardinian municipalities analyzed in this study were collected from available literature [27,28,33,34] and included the hydrogen ion concentration (pH), TDS, as well as major cations and anions, namely sodium (Na^+^), potassium (K^+^), calcium (Ca^2+^), magnesium (Mg^2+^), bicarbonate (HCO_3_^−^), chloride (Cl^−^), and sulfate (SO_4_^2^^−^). Nearly 100 municipalities out of 377 were rather considered a territorial extension of the neighboring municipalities and often encompassed a small number of inhabitants (<1000). In these cases, it was assumed that the chemical composition of the local water was similar to that of the nearest source.

### 2.4. Statistical Analysis

The mutual correlation between dissolved minerals in water was assessed by calculating the Pearson product-moment correlation coefficient. The associations between water parameters and the counts of deaths from IHD were investigated using a Bayesian conditional autoregressive (CAR) model to take into account the spatially structured random effect of neighboring areas [35], through the open-source R software (http://www.r-project.org/ (accessed on 19 May 2021)). The minerals dissolved in waters were included in the analyses as covariates. Expected case counts were calculated for mortality due to IHD (code 410–414). The total number of cases across all municipalities was divided by the total population at risk. A zero-mean normal distribution was used to model the unstructured random effects (h_i_). The regression coefficients (β_1_… k) for each of k fixed effects (x_1_… k) and for the intercept (β_0_) were assigned zero mean uninformative priors with a normal distribution.

## 3. Results

### 3.1. Correlation between Water Parameters

Table 1 displays the Pearson product-moment correlation coefficient for physico-chemical parameters of all spring waters included in this study. The pH did not show any significant correlations with the other parameters, whereas the TDS showed a significant correlation with all cations and anions except for bicarbonate and sulfate. As expected, the highest correlation was between Na^+^ and Cl^−^, followed by K^+^ and SO_4_^2^^−^. Calcium showed significant correlations with Cl^−^, Mg^2+^, and HCO_3_^−^, while Cl^−^ displayed the highest significant correlations with Na^2+^, K^+^, and SO_4_^2^^−^.

Overall, the pH of Sardinian spring waters appears skewed towards the alkaline range due to the abundance of carbonate rocks. Nonetheless, more acidic waters can be found in the North-Eastern and central area in correspondence with granites forming the mountainous backbone of the island [29,36]. The reported dissolved ions most present on average in spring waters were bicarbonate, followed by chloride and sodium. The distribution of calcium concentration and SMR from AMI and IHD in spring waters from all Sardinian municipalities is illustrated in Figure 2a–d, respectively. 

The amount of calcium showed a wide spatial variability reaching the maximum in the mineral-rich, hard waters from North-Western cape (Nurra) and South-Eastern area (Salto di Quirra), as well as in a small South-Western spot (Sulcis-Iglesiente) (Figure 2). The concentration of magnesium, related to the one of calcium, is remarkable in waters extending in a southerly direction among the sedimentary rocks of Logudoro, Marghine, and Goceano, as well as in the innermost area of Barbagia (Seulo) (Figure 2).

### 3.2. Water Minerals and Standardized Mortality Ratio for Coronary Artery Disease

In the total 377 municipalities, 9918 deaths due to CAD were retrieved from 1981–1991 period. Table 2 illustrates the Pearson correlation coefficients between the physico-chemical characteristics of Sardinian spring waters in all municipalities and the SMR for AMI and IHD. Significant negative correlations were found only for calcium (AMI: r = −0.123; IHD: r = −0.146) and borderline significance for magnesium (AMI: r = −0.131; IHD: r = −0.138) and bicarbonate (AMI: r = −0.147; IHD: r = −0.126), whereas weak positive correlations were detected for sodium and chloride. Interestingly, the pH of spring water was negatively correlated with mortality for CAD (AMI: r = −0.071; IHD: r = −0.113), suggesting a protective role of alkaline water.

The quality of the water in the three sub-regions of the island with the highest calcium content is reported in the Appendix A. Notably, in these sub-regions where in the 1980s the population consumed hard waters, the SMR for CAD was consistently and significantly below 1.0.

The average SMR for CAD in municipalities with a predominantly “soft” (<30 mg/L) or “hard” (≥30 mg/L) water was, respectively, 121.4 ± 59.1 vs. 104.7 ± 38.2 (*p* = 0.025). In other words, the areas with lower SMR for CAD are those where the spring water is richer in Ca^2+^.

This was confirmed by the subsequent spatial regression analysis, whose posterior means and 95% credible intervals for the regression coefficients are reported in Table 3.

The magnitude and sign of coefficient estimate for calcium are in accordance with the relationship illustrated in Figure 2 and highlight a significant inverse association with CAD (−0.470, *p* = 0.001). At opposite, a significant direct association with chloride was detected (0.291, *p* = 0.037). Associations with the other covariates were not significant.

## 4. Discussion

In the present study, the relationship between the physico-chemical composition of spring mineral waters and mortality for CAD in Sardinia was investigated. The availability of databases on spring water composition made it possible for the present spatial analysis.

Before the habit of drinking bottled mineral water, not necessarily of local origin, took place, most of the island population used to obtain supplies of drinking water from public fountains closest to home [25].

In general, the hardness (molar sum of Ca^2+^ and Mg^2+^) of Sardinian waters is quite variable across the island. However, the concentration of magnesium was not particularly high in Sardinian waters, despite lithogenic mechanisms and anthropic activities. In particular, magnesium was found in low concentrations in Ca^2+^-rich areas such as the Salto di Quirra.

Sodium was present in high concentrations, especially in the spring waters from the North-Western area, while potassium was mainly found in the Central-Northern area where acidic soils, rich in feldspar, predominate. Bicarbonate was the anion mostly represented in the mineral-rich waters moving along through the carbonate rocks located at the extreme ends of the island, whereas it was replaced by chloride in a few areas that underwent recent seawater intrusion [37] as well as in the low-salinity waters of the central mountain massif. Sulfate was the characteristic anion of the thermal waters flowing from eruptive rocks near once-active volcanic areas [38].

In line with previous studies, the correlation analysis showed a significant inverse association between the content of Ca^2+^ and mortality from coronary artery disease, whereas the concentration of Mg^2+^ cations showed a borderline trend in the same direction. The water salinity was not significantly associated with CAD mortality. These results were confirmed by the spatial regression analysis and at aggregate levels indicating that a high Ca^2+^ and Mg^2+^ content in spring water was strongly associated with a lower CAD SMR. The inverse relationship between water hardness and CAD was observed primarily with calcium while it was weaker with magnesium; this could be explained by the fact that in more than 150 Sardinian water sources, the magnesium content was lower than 5 mg/L.

Inhabitants of Salto di Quirra (South East of Sardinia) (Figure 2), in addition to having a lower SMR for CAD, are characterized by greater longevity, especially among the male sex [39], although, in some locations of the subregion, the water is also rich in sodium due to marine intrusion. The protective effect against CAD mortality of waters rich in calcium does not seem limited to the limestone areas of Sardinia but has been found in other long-lived areas with comparable hydrogeological characteristics: for example, in the Nicoya peninsula in Costa Rica [17,18] and Ikaria [40]. Similar results were also reported in recent studies both in Western [41,42] and in Asian [43] countries.

A trend for beneficial association was also observed for Sardinian spring waters rich in bicarbonate. For example, populations from areas such as the Coros andesitic massif (Codrongianos), where the most common drinking spring water was from the San Martino source, with a number of dissolved solids of 2808 mg/L rich in calcium content (145 mg/L) and HCO_3_^−^ anion (1305.7 mg/L) [22], showed one of the lowest SMR for CAD.

The relationship between water hardness and mortality for CAD was investigated since the 1960s [44,45]. Although several studies conducted in this period did not seem to confirm the initial hypothesis of a negative correlation, an article by Margaret Crawford, published in 1968 in the Lancet journal, corroborated the inverse relationship between water hardness and cardiovascular mortality [46]. The interest in this field was renowned by a broader perspective in geochemistry, beyond the mere effect of calcium and magnesium, extending the number of populations tested. In 2008 a systematic review of more than two thousand studies investigating the association between levels of drinking water hardness and cardiovascular disease firmly concluded for an inverse relationship with magnesium, whereas the relationship for calcium resulted unclear [47]. Our data are partially in accordance with this metanalysis and confirm the health benefit of consuming hard waters.

A number of different metabolic pathways may explain the protective effect of hard water consumption on the occurrence of CAD. The most recent research on this field underlines that the consumption of hard water is capable of preventing the pre-hypertensive state, which is one of the most important pathogenetic factors [48]. Furthermore, magnesium is a regulator of the redox balance and can improve endothelial function [49]. In addition, low levels of magnesium and calcium, i.e., low degree of hardness of the drinking water, may increase the Pb leaching from water pipes and, in turn, Pb absorption favoring hypertension [50]. The presence of HCO_3_^−^ replacing chloride increases the alkalinity of drinking water. Several studies have confirmed that drinking slightly alkaline water, close to the level in the human body and the plasma pH, can keep blood vessels more elastic, maintaining the blood pressure in the normal range levels, which is advantageous for the cardiovascular system [51].

In our study, some limitations need to be mentioned. For example, a municipality could be supplied by more than one source. In this case, it is reasonable to assume that the physico-chemical features of these multiple sources were similar based on the shared hydrogeological characteristics of soil in the same village. Moreover, it is possible that water composition may have changed with time, and thus in the decade analyzed (1981–1991). However, we have examples of Sardinian sources whose composition remained stable over more than 30 years. It could also be argued that in bigger cities, people may commute for work, potentially consuming water of different quality compared to home. However, in the decade under consideration, only 10 municipalities exceeded 20,000 inhabitants, out of a total of 1.6 million throughout the island, and commuting was not a common practice, therefore minimizing this potential bias.

## 5. Conclusions

Water is essential to sustain life, and the present ecological study suggests that minerals dissolved in spring water, and especially the balance between anions and cations, in relation to the underlying rock composition, may be of major importance for human wellness as both too much and too little mineral salts may negatively affect human health. The high concentration of calcium, far from being harmful, may be protective against disease, as suggested by numerous reports around the world. Further research is needed to investigate if hard water may also be beneficial to prevent other diseases.

## Figures and Tables

**Figure 1 nutrients-13-02858-f001:**
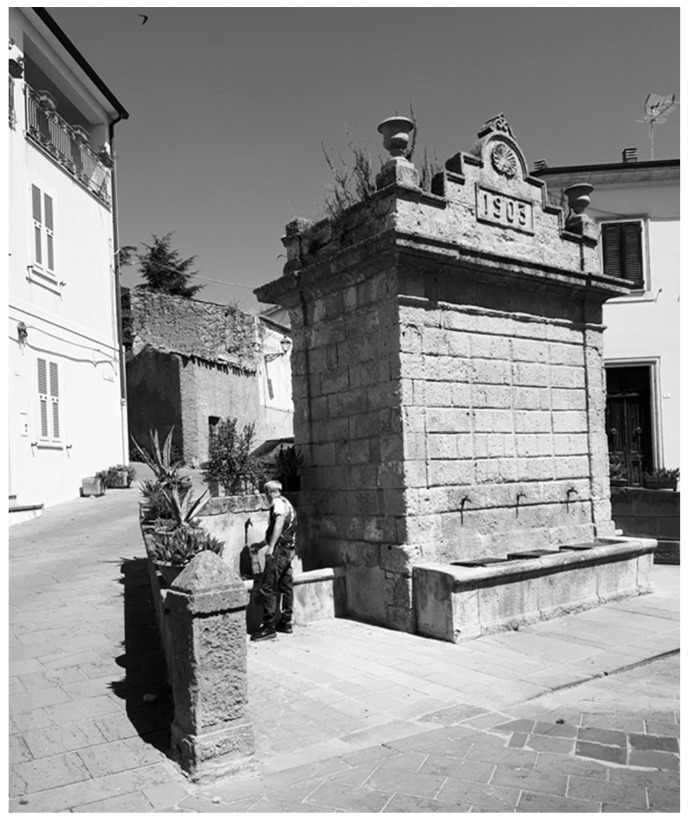
An example of a typical public fountain in a Sardinian village (Mores).

**Figure 2 nutrients-13-02858-f002:**
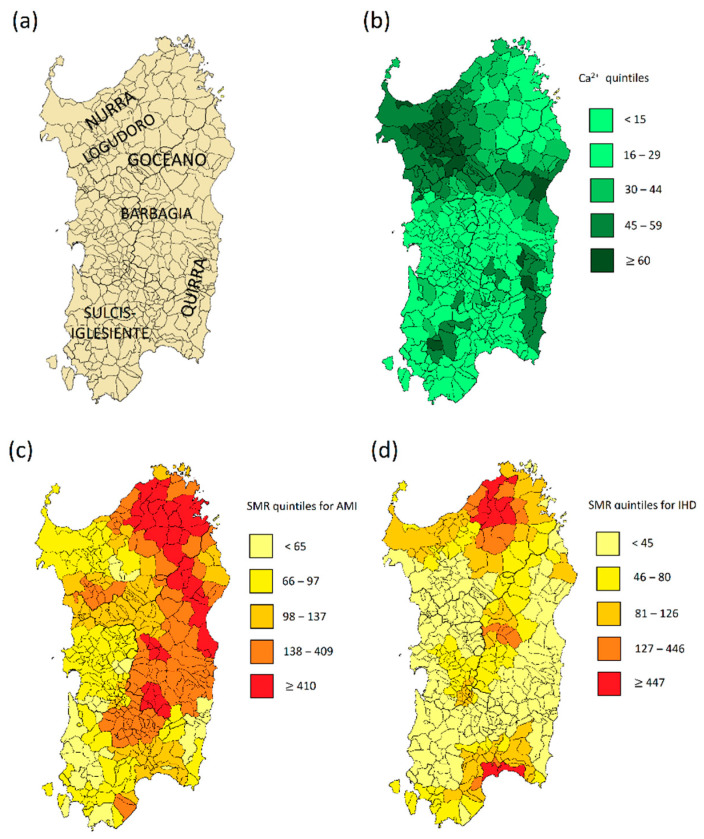
(**a**) subregions of Sardinia; (**b**) Ca^2+^ concentration in spring waters; (**c**) standardized mortality ratio (SMR) for acute myocardial infarction (AMI); (**d**) standardized mortality ratio for ischemic heart disease (IHD).

**Table 1 nutrients-13-02858-t001:** Correlation matrix between the physico-chemical characteristics of spring waters in the 377 Sardinian municipalities.

	Total Dissolved Solids	pH	Sodium (Na+)	Potassium (K+)	Calcium (Ca^2+^)	Magnesium (Mg^2+^)	Bicarbonate (HCO_3_^−^)	Sulphate (SO_4_^2^^−^)
Total dissolved solids								
pH	0.087							
Sodium (Na^+^)	0.463 **	0.322						
Potassium (K^+^)	0.363 **	−0.289	0.429 **					
Calcium (Ca^2+^)	0.286 *	−0.270	0.298 *	0.375 **				
Magnesium (Mg^2+^)	0.432 **	−0.388	0.152 *	0.132	0.234 **			
Bicarbonate (HCO_3_^−^)	0.215	−0.331	0.183 *	0.676 **	0.331 **	0.350 **		
Sulphate (SO_4_^2-^)	0.171	−0.276	0.328 **	0.834 **	0.281 **	0.191 **	0.799 **	
Chloride (Cl^−^)	0.396 **	0.211	0.844 **	0.686 **	0.334 **	0.177 *	0.439 **	0.660 **

* *p* < 0.05, ** *p* < 0.01.

**Table 2 nutrients-13-02858-t002:** Correlations between physico-chemical characteristics of Sardinian waters and mortality for acute myocardial infarction (AMI) and other ischemic heart diseases (IHD).

Parameter	Unit	Average Values in Spring Waters	Minimum and Maximum	Pearson Correlation Coefficient with Mortality for AMI	Pearson Correlation Coefficient with Mortality for IHD
Total dissolved	mg/L	529.9 ± 437.4	108–2522	−0.066	−0.057
solids					
pH		7.35 ± 0.78	6.5–9.6	−0.071	−0.113
Sodium (Na^+^)	mg/L	53.5 ± 47.2	7.4–250.0	0.036	0.074
Potassium (K^+^)	mg/L	3.6 ± 5.3	0.6–46.0	−0.032	−0.070
Calcium (Ca^2+^)	mg/L	22.6 ± 29.3	0.2–213.0	−0.123 ^1^	−0.146 ^3^
Magnesium (Mg^2+^)	mg/L	16.9 ± 28.5	0.1–345.0	−0.131 ^2^	−0.138 ^4^
Bicarbonate (HCO_3_^−^)	mg/L	164.2 ± 276.5	6.1–1159.3	−0.147	−0.126 ^5^
Sulphate (SO_4_^2-^)	mg/L	26.0 ± 36.9	1.9–203.5	−0.099	−0.121
Chloride (Cl^−^)	mg/L	84.9 ± 72.8	14.0–408.2	0.072	0.013

^1^*p* = 0.032; ^2^
*p* = 0.054; ^3^
*p* = 0.009; ^4^
*p* = 0.074; ^5^
*p* = 0.058.

**Table 3 nutrients-13-02858-t003:** Posterior means and 95% credible intervals for the regression coefficients of the spatial model involving minerals dissolved in spring water and coronary artery disease in Sardinia.

		95% CrI	
Effect	Coefficient	Lower	Upper
Intercept	113.3	100.8	125.7
Calcium (Ca^2+^)	−0.470	−0.743	−0.197
Magnesium (Mg^2+^)	−0.082	−0.228	0.065
Sodium (Na^+^)	−0.318	−0.706	0.070
Potassium (K^+^)	0.020	−3.129	3.168
Bicarbonate (HCO_3_^−^)	0.038	−0.025	0.101
Sulphate (SO_4_^2-^)	0.026	−0.671	0.724
Chloride (Cl^−^)	0.291	0.018	0.564

## Data Availability

The data presented in this study are available in the Atlante di Mortalità in Sardegna [32] and in references [27,28,33,34].

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
