# Peer review of "Water Quality and Mortality from Coronary Artery Disease in Sardinia: A Geospatial Analysis"

_nutrients, 2021, doi:10.3390/nu13082858_

Round 1

Reviewer 1 Report

Comments on the manuscript “Water quality and mortality from ischemic heart disease in Sardinia: A geospatial analysis" (author: Maria Pina Dore, Guido Parodi, Michele Portoghese, Alessandra Errigo, Giovanni Mario Pes; ref: nutrients-1303764)

General assessment:

This epidemiological study, even only of ecological nature (but authors are well aware about this limitation), represents useful contribution to our knowledge on the role of natural water minerals for human health. If the manuscript is improved, it may be published in the journal.

Specific comments:

  • Abstract, line 16. I recommend to replace word “fountain” with better term, e.g. local springs or local supplies. The fountain reminds just public fountain on the square, which could be filled in past times either from local spring or local river.
  • Abstract, line 25. You should specify, what trend (for bicarbonate) was observed. The same as for calcium or other one (?).
  • Abstract, line 27 (and throughout whole manuscript). You use term “hyper-mineral water”, which is not common and you did not define it. I strongly recommend either to avoid it or to define it clearly. According to the Table 2, total dissolved solids in Sardinian spring waters were on average about 530 mg/L, which does not provide argument to call such water(s) “hyper-mineral”. I still admit, that some of the springs could have high mineral content (e.g. TDS more than 2000-3000 mg/L) and could be called hyper-mineral, but definitely not all or most springs involved in your study. You have to define it or change in whole manuscript, not just in Abstract.
  • Page 1-2, specifically lines 43-67: I cannot agree with interpretation of current knowledge (on health role of water minerals) provided by authors in Introduction. It is clear that they have limited overview about the literature published on this issue. The authors suggest that results of the studies are more or less chaotic and in contradiction, with no clear concept, but it is not true. Of course, always you can find the single studies provided different outputs, but if you take into account all studies published, the picture is quite clear as most studies shows similar results (if you use the same study design and health criteria). And it is also natural, in case of essential elements, than very low concentration (intake) is related to some kind of pathology, some optimum concentration is related only to minimum health problems (i.e. is providing some benefits for health), and very high concentration (of the same mineral) is again related to higher incidence of some pathology (but different than that caused by very low intake). So, you have to know the health effects of each mineral and know what you can looking for. I recommend to consult some recent review on the topic, e.g. e.g. Rosborg, I., Kozisek, F. (eds.). Drinking Water Minerals and Mineral Balance. Importance, Health Significance, Safety Precautions. 2nd ed. Springer International Publishing, London 2020, DOI 10.1007/978-3-030-18034-8, ISBN-978-3-030-18033-1. Be also careful to mix the results of two kind of studies: observational epidemiological studies (looking for long term water consumption) and some experimental (clinical) studies (looking for health effect of rather short term consumption, just of some months, where – due to adaptation – you can sometimes observe contradictory, but also transient effect in comparison with long term consumption, e.g. effect of some type of sodium rich mineral water on blood pressure).
  • Page 2, line 21: the reference 21 is quite obsolent (1983), there are much more recent and better publications on the topic. And if you provide estimated threshold (of salt intake) 5 g NaCl/day to induce hypertension, you have to be aware, that very most of Western population consume this amount of salt commonly already in their diet, it is not necessary to have such intake just from water. If water has sodium 300 mg/L, it is already very important contribution to the total daily intake!
  • Page 3, Materials and methods, Data collection: You should state primarily here, what period (1981-91) was used for data selection, not (only) in 3.2.
  • Page 3, M+M, Water composition: Have you been able to find in the 4 references (28,29,33,34) really individual composition of water from each of 377 municipalities? And one municipality is supplied just from one source or two (more) sources of very similar quality? What if two (or more) sources of different composition were used? Data about water quality comes from single analyses or from some long term monitoring (how reliable it is)? Are there any bigger cities in Sardinia, where more people commute for work (and consume water of different quality than at home)? How it could influence the results? Please explain all these questions.
  • Page 3, M+M, Water composition, lines 122-123: you probably omit to mention chloride which appears in Table 1 and 2.
  • Page 4, Results, line152-153: If this occurred and you had to pool – did you pool also data on water quality from there municipalities? And after pooling, how many municipalities (units) did you include in final statistics, please?
  • Page 4, Table 2. I would expect better characterization of water quality, not just average values and (? – standard deviation, please specify), but also at least minimum and maximum values to see, what are the ranges of single parameters.
  • Page 6, Discussion, lines 197-202: according to all meta-analyses published on water Ca and Mg and cardiovascular disease, one would expect that primarily inverse relationships between magnesium and IHD is confirmed (rather than for calcium). Try to provide some explanation, why you did not find any relationships for Mg, which is surprising. Probably most municipalities had very low content (below 5 mg/L) (?), but it does not seem to be the case if looking in Table 2.
  • Page 6, Discussion, line 198: I am not aware about any limit “30 mg/L recommended by the WHO (38)” and I did not find it in any recent edition of the WHO Guidelines for Drinking-water Quality. Are you sure about the reference (which is missing the edition!)?
  • Page 6, Discussion, line 228-233: Are you sure you want to speak about chloride and not about sodium? It seems to be confused.
  • Page 6, Discussion, line 234-242: I especially do not agree with the sentence “In the following decades…”. See for the very good review of about 2 thousands studies on water hardness and cardiovascular diseases done in 2005 by University of East Anglia, which shows that the results are finally very consistent. See University of East Anglia + Drinking Water Inspectorate. Review of evidence for relationship between incidence of cardiovascular disease and water hardness. Final report. Norwich – London, 2005. (https://www.dwi.gov.uk/research/completed-research/risk-assessment-chemical/review-of-evidence-for-relationship-between-incidence-of-cardiovascular-disease-and-water-hardness/) + Catling, L.A., Abubakar, I., Lake, I.R., Swift, L., Hunter, P.R. A systematic review of analytical observational studies investigating the association between cardiovascular disease and drinking water hardness. J. Wat. Health, 2008; 6(4): 433–442.
  • Page 6, Conclusions, lines 260-261: Such studies already exist (!), but they are not so abundant as studies looking for water Ca or Mg and CVD (IHD). Look again e.g. for Rosborg, I., Kozisek, F. (eds.). Drinking Water Minerals and Mineral Balance. Importance, Health Significance, Safety Precautions. 2nd ed. Springer International Publishing, London 2020.

Author Response

Reviewer #1

Comments and Suggestions for Authors

General assessment:

This epidemiological study, even only of ecological nature (but authors are well aware about this limitation), represents useful contribution to our knowledge on the role of natural water minerals for human health. If the manuscript is improved, it may be published in the journal.

We thank the reviewer for his/her remarks. The feedback is much appreciated, and some very good points are made.

Specific comments:

Abstract, line 16. I recommend to replace word “fountain” with better term, e.g. local springs or local supplies. The fountain reminds just public fountain on the square, which could be filled in past times either from local spring or local river.

Reply: we thank the reviewer for raising this point. The word “fountain” was used exactly with the meaning specified by the reviewer. In the Sardinian villages public fountains are on the main square, fed either from local spring or local river. A new figure (figure 1), representing a typical Sardinian fountain was added to the manuscript and was shot on August 2021.

Abstract, line 25. You should specify, what trend (for bicarbonate) was observed. The same as for calcium or other one (?).

Reply: Sorry for the oversight. The mistake was fixed for bicarbonate and for calcium (lines 26-28).

Abstract, line 27 (and throughout whole manuscript). You use term “hyper-mineral water”, which is not common and you did not define it. I strongly recommend either to avoid it or to define it clearly.

According to the Table 2, total dissolved solids in Sardinian spring waters were on average about 530 mg/L, which does not provide argument to call such water(s) “hyper-mineral”. I still admit, that some of the springs could have high mineral content (e.g. TDS more than 2000-3000 mg/L) and could be called hyper-mineral, but definitely not all or most springs involved in your study. You have to define it or change in whole manuscript, not just in Abstract.

Reply: The word “hyper-mineral water” was changed throughout the manuscript and when necessary to use “hyper-mineral water” the term was defined and the reference added (page 2, lines 52-53).

Page 1-2, specifically lines 43-67: I cannot agree with interpretation of current knowledge (on health role of water minerals) provided by authors in Introduction. It is clear that they have limited overview about the literature published on this issue. The authors suggest that results of the studies are more or less chaotic and in contradiction, with no clear concept, but it is not true. Of course, always you can find the single studies provided different outputs, but if you take into account all studies published, the picture is quite clear as most studies shows similar results (if you use the same study design and health criteria). And it is also natural, in case of essential elements, than very low concentration (intake) is related to some kind of pathology, some optimum concentration is related only to minimum health problems (i.e. is providing some benefits for health), and very high concentration (of the same mineral) is again related to higher incidence of some pathology (but different than that caused by very low intake). So, you have to know the health effects of each mineral and know what you can looking for. I recommend to consult some recent review on the topic, e.g. e.g. Rosborg, I., Kozisek, F. (eds.). Drinking Water Minerals and Mineral Balance. Importance, Health Significance, Safety Precautions. 2nd ed. Springer International Publishing, London 2020, DOI 10.1007/978-3-030-18034-8, ISBN-978-3-030-18033-1.

Reply: in the revised manuscript the following sentences have been added and the suggested reference quoted: “It has been known for over a century that the mortality rate in a population is related to the hardness of water, but it was only at the end of the 1950s that the first systematic studies aimed at clarifying this relationship were conducted [3]. ……………………For many years, the prevailing opinion was that consumption of mineral-rich waters, defined as TDS exceeding the threshold of 1500 mg/L [4]” (page 2, lines 46-54).

Be also careful to mix the results of two kind of studies: observational epidemiological studies (looking for long term water consumption) and some experimental (clinical) studies (looking for health effect of rather short term consumption, just of some months, where – due to adaptation – you can sometimes observe contradictory, but also transient effect in comparison with long term consumption, e.g. effect of some type of sodium rich mineral water on blood pressure).

Reply:  in the revised manuscript the literature in the field has been presented in a more coherent way. “More specifically, sodium content in water was considered deleterious for its potential to induce blood hypertension, although chloride concentration may contribute as well [8,14-16]. Nonetheless, other studies seem to indicate beneficial outcomes from mineral-rich water consumption, including life-span extension in long-lived populations [17,18]. In reality, as far as the relationship between mineral-rich water and cardiovascular risk is concerned, although short-term clinical studies may display transient adverse effects ‒ i.e., a pressure-raising effect ‒ in comparison with epidemiological studies entailing long term water consumption, accumulating evidence consistently indicate that regular consumption of waters rich in calcium and magnesium, provides health benefit, especially in terms of reducing cardiovascular disease burden [3,19].” (page 2, lines 57-67)

Page 2, line 21: the reference 21 is quite obsolent (1983), there are much more recent and better publications on the topic.

Reply: that reference was suppressed.

And if you provide estimated threshold (of salt intake) 5 g NaCl/day to induce hypertension, you have to be aware, that very most of Western population consume this amount of salt commonly already in their diet, it is not necessary to have such intake just from water. If water has sodium 300 mg/L, it is already very important contribution to the total daily intake!

Reply: all this information was suppressed in the revised manuscript.

Page 3, Materials and methods, Data collection: You should state primarily here, what period (1981-91) was used for data selection, not (only) in 3.2.

Reply: The period was added in the data collection section (page 4, line 117-118).

Page 3, M+M, Water composition: Have you been able to find in the 4 references (28,29,33,34) really individual composition of water from each of 377 municipalities?

And one municipality is supplied just from one source or two (more) sources of very similar quality? What if two (or more) sources of different composition were used? Data about water quality comes from single analyses or from some long term monitoring (how reliable it is)? Are there any bigger cities in Sardinia, where more people commute for work (and consume water of different quality than at home)? How it could influence the results? Please explain all these questions.

Reply: We thank the reviewer for raising these important issues. From all the available literature reporting data on the chemical composition of waters in Sardinia, we retrieved data relating to sources of about 250 municipalities, the remaining municipalities being rather a territorial extension of the former, often accounting a small number of inhabitants (<1000). In these cases, it was assumed that the chemical composition of the local water was similar to that of the nearest source in a sampled municipality. In doing this, the potential discrepancy was considered negligible. In larger municipalities there was sometimes more than one spring source, in which case the composition of the main source was selected for analysis (very often the one for which the only data were available). Even if now the majority of sources in Sardinia are continuously monitored, we selected only data available in the decade 1981-91, corresponding to those for which mortality data were available. In regards of the impact of commuting in water consumption, in the decade under consideration, only 10 municipalities exceeded 20,000 inhabitants, out of a total of 1.6 million throughout the island, therefore we judged this factor almost irrelevant, in addition to the fact that commuting in Sardinia is not a common practice. In the revised manuscript a subsection of limitations was added: page 9, lines 282-292.

Page 3, M+M, Water composition, lines 122-123: you probably omit to mention chloride which appears in Table 1 and 2.

Reply: Sorry, in the revised manuscript the chloride was added (page 4, line 133).

Page 4, Results, line152-153: If this occurred and you had to pool – did you pool also data on water quality from there municipalities? And after pooling, how many municipalities (units) did you include in final statistics, please?

Reply: as specified above, we have performed a pooling of both mortality and water quality data, and these combined unifications resulted in an effective number of 250 statistical units on the total of 377 municipalities. Page 4, lines 133-136

Page 4, Table 2. I would expect better characterization of water quality, not just average values and (? – standard deviation, please specify), but also at least minimum and maximum values to see, what are the ranges of single parameters.

Reply: a column with minimum and maximum values was added to the table 2 (page 7).

Page 6, Discussion, lines 197-202: according to all meta-analyses published on water Ca and Mg and cardiovascular disease, one would expect that primarily inverse relationships between magnesium and IHD is confirmed (rather than for calcium). Try to provide some explanation, why you did not find any relationships for Mg, which is surprising. Probably most municipalities had very low content (below 5 mg/L) (?), but it does not seem to be the case if looking in Table 2.

Reply: In the revised table 2 it is evident that in more than 150 sources of Sardinia municipalities the Mg content was lower than 5 mg/L. Furthermore, we added the correlation with mortality from acute myocardial infarction, obtaining the same result: an inverse relationship which, however, does not reach the threshold of statistical significance. Page 8, lines 240-243.

Page 6, Discussion, line 198: I am not aware about any limit “30 mg/L recommended by the WHO (38)” and I did not find it in any recent edition of the WHO Guidelines for Drinking-water Quality. Are you sure about the reference (which is missing the edition!)?

Reply: Thanks for your observation, this reference which had originated from a misunderstanding with the average quantity of water was eliminated

Page 6, Discussion, line 228-233: Are you sure you want to speak about chloride and not about sodium? It seems to be confused.

Reply: We agree with the reviewer that it seems confusing and this paragraph was suppressed.

Page 6, Discussion, line 234-242: I especially do not agree with the sentence “In the following decades…”. See for the very good review of about 2 thousand studies on water hardness and cardiovascular diseases done in 2005 by University of East Anglia, which shows that the results are finally very consistent. See University of East Anglia + Drinking Water Inspectorate. Review of evidence for relationship between incidence of cardiovascular disease and water hardness. Final report. Norwich – London, 2005. (https://www.dwi.gov.uk/research/completed-research/risk-assessment-chemical/review-of-evidence-for-relationship-between-incidence-of-cardiovascular-disease-and-water-hardness/) + Catling, L.A., Abubakar, I., Lake, I.R., Swift, L., Hunter, P.R. A systematic review of analytical observational studies investigating the association between cardiovascular disease and drinking water hardness. J. Wat. Health, 2008; 6(4): 433–442.

Reply: thanks for your suggestions. The paragraph was deeply reshaped. Page 9, lines 264-269.

Page 6, Conclusions, lines 260-261: Such studies already exist (!), but they are not so abundant as studies looking for water Ca or Mg and CVD (IHD). Look again e.g. for Rosborg, I., Kozisek, F. (eds.). Drinking Water Minerals and Mineral Balance. Importance, Health Significance, Safety Precautions. 2nd ed. Springer International Publishing, London 2020.

Reply: Thanks again for your precious advices based on your expertise. Conclusions were changed in the revised manuscript accordingly, and the suggested reference quoted.

Reviewer 2 Report

The role of water hardness on human health is still debated, ranging from beneficial to harmful. In this ecological study, in Sardinia, Italy, the spatial association between spring water quality/composition and standardized mortality ratio for ischemic heart disease, in the decade from 1981-1991, was investigated using data retrieved from published databases. The results, within the limitation of the ecological study, suggest a beneficial role of waters with high content in calcium and bicarbonate against ischemic heart disease.

The study contributes to the debate about the effect of the high concentration of water calcium on the disease. The conclusions of the study reveal that the hard water maybe protective against disease. The limitation of the study is that is retrospective.

Author Response

Reply: thanks a lot for the appreciation on our study

Reviewer 3 Report

The aim of the study by Pina Dore et al. was an analysis of the relationship between spring water quality and IHD mortality in Sardinia. The results of the study suggest a beneficial role for waters with high calcium and bicarbonate content against IHD.

The article is well planned, well conducted and very well described.

I only suggest supplementing it with the following issues:

  1. In the introduction, some specific epidemiological data on IHD and its complications should be given to better substantiate the importance of this disease.
  2. Basic demographic data of the studied group should be added to the material and methods section.
  3. In the regression analysis, factors that potentially model the analyzed relationships should be taken into account.
  4. Both in the introduction and in the discussion, older references should be avoided. It is now good practice to cite articles from the last 20 years.

Author Response

The aim of the study by Pina Dore et al. was an analysis of the relationship between spring water quality and IHD mortality in Sardinia. The results of the study suggest a beneficial role for waters with high calcium and bicarbonate content against IHD.

The article is well planned, well conducted and very well described.

I only suggest supplementing it with the following issues:

  1. In the introduction, some specific epidemiological data on IHD and its complications should be given to better substantiate the importance of this disease.

Reply: in the revised manuscript a short mention of IHD epidemiology was added (page 2, lines 49-52).

  1. Basic demographic data of the studied group should be added to the material and methods section.

Reply: in the revised manuscript basic demographic data on Sardinia population was added  (page 3, lines 96-99; Page 4, lines 111-114).

  1. In the regression analysis, factors that potentially model the analyzed relationships should be taken into account.

Reply: unfortunately these data were unavailable.

  1. Both in the introduction and in the discussion, older references should be avoided. It is now good practice to cite articles from the last 20 years.

Reply: we would agree with the reviewer's opinion, however the manuscript is based on historical data and to use old reference was “a conditio sine qua non”. However, as far as possible, in the revised version of the manuscript we have attempted to limit these documentary sources.

Reviewer 4 Report

In the article of Dore et al. entitled „ Water quality and mortality from ischemic heart disease in Sardinia: A geospatial analysis” Authors developed retrospective study of water quality impact on the mortality rate for ischemic heart disease in the Sardinia Island population. Overall impression is that the paper is well written, with proper use of citations. I suggest this manuscript can be accepted for publication in Nutrients after minor revision: 1. Please add abbreviation for IHD also in manuscript, not only in the abstract. 2. As I understand it, the data in the Table 1 are the average for all regions, but later in the text the Authors describe them as specific parts of the Island (North-Eastern, Central-Northern etc.), so please add relevant statistics for these regions, if such data are available. 3. If relevant data are available, please compare the mortality rate for IHD after the change of the consumption habits to bottled water.

Author Response

In the article of Dore et al. entitled „ Water quality and mortality from ischemic heart disease in Sardinia: A geospatial analysis” Authors developed retrospective study of water quality impact on the mortality rate for ischemic heart disease in the Sardinia Island population. Overall impression is that the paper is well written, with proper use of citations. I suggest this manuscript can be accepted for publication in Nutrients after minor revision:

  1. Please add abbreviation for IHD also in manuscript, not only in the abstract.

Reply: In the revised manuscript we tried to be more detailed and the coronary artery disease (CAD) was use to refer to both acute myocardial infarction (AMI) and ischemic heart disease (IHD. More attention was paid in the use of abbreviation throughout the manuscript.

As I understand it, the data in the Table 1 are the average for all regions, but later in the text the Authors describe them as specific parts of the Island (North-Eastern, Central-Northern etc.), so please add relevant statistics for these regions, if such data are available.

Reply: please see the supplemental material.

  1. If relevant data are available, please compare the mortality rate for IHD after the change of the consumption habits to bottled water.

Reply: unfortunately these data were unavailable.

Round 2

Reviewer 3 Report

The authors responded to my suggestions. The manuscript has been partially revised. The failure to take into account modulating factors in the regression analysis is a significant limitation of the study. This aspect needs to be mentioned and commented upon when discussing the limitations of the manuscript.